# The Association of Personality Traits and Parameters of Glycemic Regulation in Type 1 Diabetes Mellitus Patients Using isCGM

**DOI:** 10.3390/healthcare10091792

**Published:** 2022-09-17

**Authors:** Daria Sladić Rimac, Ines Bilić Ćurčić, Ivana Prpić Križevac, Ema Schonberger, Maja Gradinjan Centner, Marija Barišić, Silvija Canecki Varžić

**Affiliations:** 1Department of Endocrinology and Metabolism Disorders, Clinical Hospital Center Osijek, 31000 Osijek, Croatia; 2Faculty of Medicine, University J.J.Strossmayer, 31000 Osijek, Croatia; 3Faculty of Dental Medicine and Health Osijek, Department of Nursing and Palliative Medicine, University J.J.Strossmayer, 31000 Osijek, Croatia

**Keywords:** blood glucose self-monitoring, isCGM, type 1 diabetes mellitus, personality traits

## Abstract

This study aimed to examine the impact of personality on glycemic regulation in adult patients with type 1 diabetes mellitus (T1DM). The study group consisted of subjects with T1DM, who were ≥ 18 years of age. The study was conducted in two phases: At baseline, subjects completed the Croatian version of the International Personality Item Pool scale (IPIP50s) and a questionnaire designed to gather socioeconomic data, duration of diabetes, presence of chronic complications, presence of cardiovascular risk factors, frequency, and type of pre-existing hypoglycemic episodes per week. Blood and urine samples were collected and body mass index (BMI) was calculated. Each participant was provided with the intermittently scanned glucose monitoring system (isCGM) Freestyle Libre. During the second visit (3 months from the start of the trial), glycemic parameters were collected from the reports generated from the Freestyle Libre system. Estimated glycated hemoglobin (HbA1c) values were significantly lower after three months compared to baseline HbA1c (Wilcoxon test, *p* < 0.001). An inverse correlation between the number of daily scans and degree of extraversion among subjects was observed, e.g., higher degrees of extraversion resulted in lower numbers of daily scans, while lower degrees of extraversion, i.e., introvertedness, resulted in higher numbers of daily scans (*Rho* = −0.238 *p* = 0.009). There was a positive correlation between emotional stability and time spent in hypoglycemia (*Rho* = 0.214; *p* = 0.02). In addition, a shorter duration of diabetes was associated with higher percentages of TIR and vice versa (*p* = 0.02). Investigating personality traits can be a useful tool for identifying patients predisposed to hypoglycemia and lower scanning frequency. Patients with a longer history of T1DM require closer follow-up and should be re-educated when necessary.

## 1. Introduction

Diabetes mellitus type 1 (T1DM) is a chronic metabolic condition caused by the autoimmune destruction of pancreatic β-cells resulting in absolute insulin deficiency [1]. Treatment with intensive insulin therapy for the management of T1DM was established as the standard of care based on the results of the Diabetes Control and Complication Trial (DCCT) [2] and can be administered with multiple daily injections (MDI) or an insulin pump (CSII). Glycated hemoglobin (HbA1c) is currently the most widely used measure for assessing glycemic control and the risk of long-term diabetes complications in T1DM patients. However, HbA1c used alone has several limitations and may be insufficient to optimally guide personalized changes in therapy as it does not address acute glycemic excursions, the duration, and timing of hypo and hyperglycemia, or the presence of glycemic variability [3]. Self-monitoring of blood glucose (SMBG), based on capillary glucose testing, remains the longest-used method for monitoring glucose levels and maintaining glycemic control in insulin-treated DM patients [4]. The requirement to perform a fingerpick to obtain a blood sample can be time-consuming, inconvenient, and painful for many patients consequently leading to poor compliance and limitations of the potential benefits of SMBG [5]. Over the past decade, personal continuous glucose monitoring (CGM) has become the new standard of care for numerous patients with diabetes [6]. CGM provides the user with information about the current glucose level, direction, and rate of change as well as previous glucose trends and patterns. Analysis of CGM data by either the user or clinician provides a more complete picture of glycemic patterns throughout the day and night, including time in range and the degree of glycemic variability. Three types of CGM systems are currently available: real-time CGM (rtCGM), intermittently viewed CGM (ivCGM), and intermittently scanned CGM (isCGM) [7], which were used in this study. The isCGM system provides the same type of glucose data measured with rtCGM but requires the user to purposely scan the sensor to obtain information and does not have alerts and alarms [8]. Numerous studies show that flash glucose monitoring is associated with significant improvements in HbA1c in adults and children with T1DM [9,10,11]. Despite the recent widespread availability of modern diabetes monitoring technology, data demonstrates that only a minority of adults and youth with T1D in the United States and Europe achieve goals for HbA1c [12,13].

For understanding and predicting human behavior in different situations, personality traits are of special importance [14]. The Big Five dimensions are the most widely used and well-established system in psychological science for organizing personality traits [15]. The role of personality factors has been studied as one of the psychological aspects of DM [16], and much evidence can be found in the scientific literature regarding their influence on self-care and T1DM outcomes. Some studies have demonstrated an association between personality traits and HbA1c levels; for instance, higher conscientiousness and agreeableness were linked to lower and more stable HbA1c [17,18], while lower openness was associated with higher HbA1c levels although the other four traits showed no correlation [19]. Other studies have found no association between personality and HbA1c among individuals with diabetes [20].

The current literature does not provide definite answers on whether and to what extent character affects diabetes management, but all previous studies were conducted on people using glucometers and SMBG, complicated and time-consuming methods affecting the quality of life. However, there are no studies investigating whether there is a general relationship between personality traits and glycemic management in patients using isCGM. We hypothesized that glycemic control would not be affected by personality traits to such an extent because of the use of advanced and user-friendly technology, such as CGM systems. Therefore, this study aimed to examine the possible link between personality traits and parameters of glycemic regulation in adult patients with T1DM using iCGM.

## 2. Materials and Methods

### 2.1. Subjects

Subjects were recruited from the Endocrinology Department of University Hospital Centre Osijek. The study group consisted of 155 subjects with T1DM, treated with MDI or CSII. Inclusion criteria were: individuals ≥ 18 years of age, diabetes type 1 diagnosis at least 12 months prior to the study inclusion, and CGM naive. Exclusion criteria were as follows: type 2 diabetes, specific types of diabetes due to other causes, pregnant women, history of allergic contact dermatitis on isobornyl acrylate, administration of drugs that affect glycemic status (corticosteroids), and acute psychiatric treatment.

### 2.2. Study Tools

A questionnaire prepared by the authors was used to gather data on educational level, age, sex, duration of diabetes, presence of chronic complications (retinopathy, neuropathy), weekly physical activity (hours per week), presence of cardiovascular risk factors (hypertension, active smoking), frequency and type of pre-existing hypoglycemic episodes per week, and socioeconomic status.

Venous blood samples were taken for the measurement of HbA1c, lipid metabolism (total cholesterol, HDL and LDL of cholesterol, triglycerides), and urine samples were taken for the measurement of microalbuminuria. All samples were analyzed using standardized laboratory techniques.

The Big Five personality dimensions were measured using the validated Croatian version of the International Personality Item Pool scale (IPIP50s) [21], created by Goldberg [22]. The IPIP50s consists of 50 items (10 items per factor), with each item having two opposing anchor statements aimed at assessing trait variability on the personality dimensions. The subject responds on a multiple-choice Likert-type scale with five answer options ranging from very inaccurate to very accurate. Reliability estimates (coefficient alpha) for the whole scale were 0.878, which were similar to the IPIP50s’ reliability estimates in studies with different participant samples in Croatia [21]. Each domain is expressed with a score from 10 to 50. The higher value indicates more profoundly expressed dimensions, e.g., a high score for extraversion reflects high levels of extrovertedness, whereas a low score reflects low levels of extrovertedness, i.e., introvertedness. This scoring scale was used to assess each personality dimension; hence, high scores indicate a more profound expression of a personality trait, and a low score indicates a less profound expression.

At baseline, in accordance with the American Diabetes Association guidelines [23], participants were divided using an HbA1c cut-off of ≤7.0%, indicating good blood glucose control, and HbA1c > 7.0%, indicating poor blood glucose control, and according to time in range (TIR), participants were divided using a cut-off point of >70%, indicating good blood glucose control, and subjects with TIR < 70% were classified as having poor blood glucose control [24]. Parameters of glycemic control accepted by international consensus guidelines for interpreting continuous glucose monitoring (mean glucose, percentage of estimated HbA1c, percentage of time in target range (TIR, 3.9–10.0 mmol/L, ≥70%), percentage of time above the target range (TAR, >10.1 mmol/L), percentage of time below target range (TBR, >3.9 mmol/L), scanning frequency, percentage of captured isCGM data, and several hypoglycemic episodes [24] were measured and analyzed using reports generated from the FreeStyle Libre flash glucose monitoring system (Abbott, Chicago, IL, USA).

All subjects were enrolled in a structured educational program using a standardized protocol which was conducted by trained diabetes healthcare professionals. The structured education entailed individual educational workshops for a duration of 45 min or more as needed before initiating the use of the new technology for glucose monitoring. Subjects were given general information to understand how the Freestyle Libre system functions. Participants were instructed to confirm the blood glucose level with SMBG measurement in the case of low (<3.9 mmol/L) or high (>13.9 mmol/L) glucose readings, rapidly changing glucose levels as indicated by upward or downward arrows next to the readings, or when readings did not match their clinical symptoms. Furthermore, subjects were instructed to scan the sensor at least once every eight hours to avoid data omission.

Body mass index (BMI) was calculated and categorized as normal <25 kg/m^2^ or overweight ≥25 kg/m^2^.

### 2.3. Study Protocol

During the first visit, subjects completed questionnaires and individually completed the structured educational workshop. Healthcare professionals collected blood and urine samples and analyzed body composition and BMI. The isCGM was applied to each participant. During the second visit (3 months from the start of the trial) data was collected from the reports generated by the Freestyle Libre system.

### 2.4. Statistical Analysis

The categorical data were summarized using absolute and relative frequencies and the categorical variables were tested using the chi-squared test (*χ*^2^). A Shapiro-Wilk test was performed to show the normal distribution of the numerical variables, which showed that the distribution of the variables departed significantly from normality. Based on this outcome, a non-parametric test was used, and the median with the interquartile range was used to summarize the variable. To test for the continuous variable between two independent groups we used the Mann-Whitney U test (95% CI), and to measure the differences between measurements we used the Wilcoxon test (95% CI). The measure of rank correlation is shown using Spearman’s rank correlation coefficient. The assessment of the internal consistency of the set of scales is shown using the Cronbach alpha coefficient. All *p*-values are two-sided, and the significance level was set at Alpha = 0.05. Data analysis was conducted using the statistical program MedCalc^®^ Statistical Software version 20.100 (MedCalc Software Ltd., Ostend, Belgium; https://www.medcalc.org; 2022.) and IBM SPSS 23 (IBM Corp. Released 2015. Armonk, NY, USA.

## 3. Results

The study included 155 participants with T1DM, 95 females and 60 males aged 18–76 years. Most participants, 128 (82.6%), used MDI, while 27 (17.4%) were treated with CSII. Baseline characteristics of the group are presented in Table 1 and Table 2.

Median of duration DM was 14 years (interquartile range 6–21), while median of baseline HbA1c was 7.9% (interquartile range 6.9–8.9). Self-reported baseline symptomatic hypoglycemic events were present in 137 subjects (88.4%), while self-reported asymptomatic hypoglycemic events were present in 56 (36.1%) subjects (Table 3).

At the follow-up visit, median mean glucose was 8.95 mmol/L (interquartile range 7.6–10.8), the percentage of estimated HbA1c was 7.2%, while isCGM data capture was high, 93% (interquartile range 83–97.5) (Table 4).

Estimated HbA1c values were significantly lower after three months compared to baseline HbA1c (Wilcoxon test, *p* < 0.001) (Table 5).

There were no significant differences in glycemic control among subjects with varying levels of education (data not shown).

According to baseline HbA1c, glycemic control was significantly lower among subjects with low-income levels (*p* = 0.04) (Table 6).

Differences between parameters of glycemic regulation and BMI were not registered (Table 7).

The Cronbach alpha reliability coefficient for internal consistency of the entire scale is 0.878, indicating that the scale is a reliable tool for assessing the five personality dimensions in our sample (Table 8).

Considering self-reporting of the incidence and frequency of symptomatic hypoglycemic events, no differences were observed concerning the spectrum of personality dimensions (data not shown).

Individuals with asymptomatic hypoglycemic events had a statistically higher degree of agreeableness than those without asymptomatic hypoglycemic events (*p* = 0.03) (Table 9).

No differences in personality dimensions were observed between individuals with self-reported symptomatic and asymptomatic incidences of hypoglycemia (data not shown). 

The relationship between the domain of personality traits and glycemic measurements obtained from Freestyle Libre was assessed utilizing Spearman’s rank correlation coefficient. An inverse, i.e., negative, correlation between the number of daily scans and degree of extraversion among subjects was observed, e.g., higher degrees of extraversion, i.e., extrovertedness, resulted in lower numbers of daily scans, while lower degrees of extraversion, i.e., introvertedness, resulted in higher numbers of daily scans (*Rho* = −0.238; *p* = 0.009). There was a positive correlation between emotional stability and time spent in hypoglycemia (*Rho* = 0.214; *p* = 0.02). The inverse correlation between age and three personality dimensions (extraversion, emotional stability, and intellect) was statistically significant (Table 10).

A lower level of intellect was associated with the development of chronic complications (*p* = 0.02) (Table 11).

Differences in personality dimensions in the presence of nephropathy and retinopathy occurrences were not significant (data not shown). Subjects with polyneuropathy had a lower degree of intellect than those without polyneuropathy (*p* < 0.001) (Table 12).

The relationship between baseline HbA1c, estimated HbA1c, TIR, and the duration of diabetes was assessed utilizing Spearman’s rank correlation coefficient. A shorter duration of diabetes was associated with higher percentages of TIR and vice versa (*p* = 0.02) (Table 13).

## 4. Discussion

Results of this study demonstrated significant improvement in glycemic management three months from isCGM initiation, which is consistent with clinical experience to date [25,26,27,28]. Significantly higher HbA1c measurements at baseline were seen in subjects with lower socioeconomic status; these results have also previously been seen in adults and children with T1DM [29,30]. Interestingly, no differences in HbA1c among individuals, regardless of socioeconomic status, were apparent after usage of isCGM, despite assumptions that individuals with lower socioeconomic status have lower levels of educational attainment, thus lower comprehension of medical advice and lower adoption of desirable health behaviors. Tan et al. explained that glycemic regulation using TIR guidelines is more easily understood by patients in comparison to using HbA1c [31], which may be the underlying reason that subjects with lower socioeconomic status achieve better glycemic regulation when using isCGM. Accordingly, individuals of lower socioeconomic status could channel more attention to existential-related concerns than to good health behavior.

Although T1DM was traditionally considered a disease of normal-weight individuals, obesity affects an increasing number of patients suffering from T1DM, the prevalence rate ranging between 2.8% and 37.1% [32]. High prevalence was also observed in this study (33.5%), but there was no difference in the parameters of glycemic regulation according to nutritional status. This could be attributed to the fact that probably diabetes management in type 1 diabetes largely depends on performing regular blood glucose monitoring using glucometers or CGM systems and adequate administration of insulin therapy, unlike type 2 diabetes.

In this study, subjects with high levels of agreeableness more often had self-reported asymptomatic hypoglycemic events. Previous research indicated that subjects with higher agreeableness had lower HbA1c [17,33], due to a tendency for compromise and avoidance of conflict, which in the context of T1DM propels individuals to strive for better glycemic regulation and adherence to target levels according to guidelines. Consequently, this behavior may lead to a higher risk of hypoglycemic episodes. This significance was no longer evident after isCGM use, as was shown in a study with 10,370 diabetic patients using isCGM, of which 80% reported that isCGM usage reduced the proportion of time spent in hypoglycemia [34]. Agreeableness is a socially desirable personality dimension; however, our results have shown that individuals with higher levels of agreeableness characterized by attentiveness to others are usually diverted from focusing on their health condition. Clinical trials in children and adolescents found that personality traits are correlated to glycemic control assessed via HbA1c [17,33], whereas our results showed a lack of correlation among adult subjects with T1DM, which is also evident in several previous clinical trials with T1DM and T2DM patients [20,35,36,37].

It is important to note that only a small number of clinical trials have assessed personality traits among adult patients with T1DM to date, in which HbA1c was the parameter used for monitoring glycemic regulation. In this study, no association between differences in personality traits and estimated HbA1c, BMI, or TIR was found. To our knowledge, this is the first study assessing the relationship between personality traits and CGM metrics.

Our results demonstrated a significant relationship between emotional stability and the percentage of TBR, whereas a previous trial conducted among adult subjects treated with insulin pumps showed that higher conscientiousness may be related to more frequent hypoglycemic episodes [37]. On the contrary, three-year monitoring of children and adolescents with T1DM linked high and low levels of emotional stability with poor glycemic control assessed with HbA1c [17].

Despite emphasizing the need for frequent scanning during structural education, typically a predictor for lower HbA1c [34], subjects with higher extraversion ultimately had lower scanning frequency per day. A study investigating the influence of personality on mHealth app use in patients with diabetes suggested that extroverted diabetic patients were less likely to adopt the mHealth app, even though it incorporated and emphasized social features. This may be explained by the fact that the study focused on motivating long-term and continuous self-management, which may not fulfill the social desires of extroverts [38]. Over the years, it has been widely accepted that extraversion in individuals is more desirable and positive than introversion. Positive aspects of extraversion contribute to stable social interactions and support systems, subsequently protecting highly extroverted individuals from the undesirable effects of loneliness. Negative aspects of extraversion are attributed to increased excitement-seeking activities for stimulation which may more frequently lead to unhealthy behavior, for example, frequent smoking and alcohol consumption [39,40]. In the context of our research, it could be hypothesized that extraverted individuals focus more on social activities, consequently neglecting health recommendations.

In addition, age was inversely and significantly correlated to extraversion, emotional stability, and intellect, which is in agreement with previous studies implying a positive association of age with levels of agreeableness and conscientiousness, whereas levels of extraversion and intellect are negatively, or inversely, associated with age [41]. It is possible that with aging, the expression of personal traits relating to social interest and community increases whereas the expression of personal traits relating to curiosity and creativity declines. Of relevance in this study is the finding that the expression of intellect was significantly less expressed in subjects with a long duration of diabetes and chronic complications. In addition, polyneuropathy was more evident in subjects with lower intellect scores. Because intellect implies openness to challenges [42] and new experiences, it can be presumed that individuals with chronic complications focus more on health issues, consequently leading to a diminished desire for new experiences.

The duration of diabetes among subjects in this study has been shown to be a predictor of lower percentages of TIR as was shown in a study conducted by Petricic et al. One could argue that individuals with diabetes become increasingly more conflict-avoidant as time passes, increasingly distanced from addressing health problems, and ultimately more prone to poor glycemic control [43] and the development of chronic complications.

Although the number of subjects included in our study was relatively large, the results obtained originate from one hospital center in Croatia, which may be a limiting factor for projecting outcomes for all of Croatia or worldwide. In addition, the follow-up period was rather short. A long-term multicenter study including other variables such as quality of life, knowledge level, and attitude toward diabetes should be considered for future prospective trials. However, presently, there are no studies published assessing the influence of personality traits in isCGM users with T1DM.

## 5. Conclusions

The use of CMG demonstrated a significant improvement in glycemic regulation among adults with T1DM during the period of three months. We have demonstrated for the first time that determining personality traits could be useful for identifying patients using isCGM predisposed to hypoglycemia and lower scanning frequency. Patients with a longer history of T1DM require closer follow-up and should be re-educated when necessary. In the future, modern isCGM devices will not require scanning and will have alerts indicating hypo- or hyperglycemia, helping patients to maintain safer glycemic control, possibly avoiding the influence of personality traits completely or to a greater extent on diabetes management. Furthermore, the results obtained in this study could be implemented in the design of educational programs for diabetic patients. Still, further longitudinal studies are required to determine whether personality traits impact glycemic regulation in adult patients with T1DM.

## Figures and Tables

**Table 1 healthcare-10-01792-t001:** Baseline characteristics of subjects participating in the study.

Sex [*n*(%)]	
Male	60 (38.7)
Female	95 (61.3)
Marriage status [*n*(%)]	
Unmarried/divorced	65 (41.9)
Married	90 (58.1)
Place of residence [*n*(%)]	
Urban	86 (55.5)
Rural	69 (44.5)
Level of education [*n*(%)]	
Unfinished primary school/elementary school/high school	115 (64.2)
Higher education	40 (25.8)
Work status [*n*(%)]	
Unemployed	44 (28.4)
Employed	94 (60.6)
Retired	17 (11)
Income level [*n*(%)]	
<200 E	18 (11.6)
200–600 E	64 (41.3)
>600 E	73 (47.1)
Smoking habit [*n*(%)]	
No	72 (46.5)
Yes	51 (32.9)
Quit	32 (20.6)
Habitual alcohol consumption	
No—less than 1× per week	108 (69.7)
Moderate—1 dL wine up to 2× per week	8 (5.2)
Yes—1 dL wine every day or up to 2× per week	4 (2.6)
Physical activity frequency	
None	19 (12.3)
Light activity (walking, bicycling)	123 (79.4)
Regular activity (intensive)	13 (8.4)
Changes in body weight	
None	86 (55.5)
Yes, weight loss	41 (26.5)
Yes, weight gain	28 (18.1)
Insulin administration	
CSII	27 (17.4)
MDI	128 (82.6)

Euro (E, currency of the European Union), multiple daily injections (MDI), continuous subcutaneous insulin infusion (CSII).

**Table 2 healthcare-10-01792-t002:** Body measurements and composition.

	Median (IQR)	Minimum–Maximum
Age (years)	38 (27.75–48.25)	18–76
Weight (kg)	69.6 (60.6–83)	42.4–128.3
Height (cm)	172 (164–180)	150–203
BMI (kg/m^2^)	23.5 (21.5–26.4)	15–41.6
Nutritional status [(%)]		
Normal	103 (66.5%)	
Overweight	52 (33.5%)	
BMI (kcal)	1510 (1283–1793)	992–2494

Body mass index (BMI).

**Table 3 healthcare-10-01792-t003:** Parameters of glycemic control at baseline.

		Minimum–Maximum
Duration of diabetes (years) [Median (IQR)]	14 (6–21)	0–51
Insulin [Median (IQR)]	40 (32–50)	11–70
HbA1c (*n* = 152) [Median (IQR)]	7.9 (6.9–8.9)	4.9–12.7
SMBG per day [Median (IQR)]	5 (4–6)	2–18
Symptomatic hypoglycemic events [*n*(%)]	137 (88.4)	
Symptomatic hypoglycemic events [*n*(%)]		
Nocturnal	28 (24.6)	
Daily	40 (35.1)	
Both	46 (40.4)	
Frequency of nocturnal hypoglycemic events [*n*(%)]		
Less than 1× per week	36 (30.5)	
More than 1× per week	82 (69.5)	
Asymptomatic hypoglycemic events [*n*(%)]	56 (36.1)	
Asymptomatic hypoglycemic events [*n*(%)]		
Nocturnal	11 (25)	
Daily	24 (54.5)	
Both	9 (20.5)	
Frequency of asymptomatic hypoglycemic events [*n*(%)]		
Less than 1× per week	28 (62.2)	
More than 1× per week	17 (37.8)	

Glycated hemoglobin (HbA1c), self-monitoring of blood glucose (SMBG).

**Table 4 healthcare-10-01792-t004:** Parameters of isCGM glycemic control after 3 months.

isCGM	Median (IQR)	Minimum–Maximum
Mean glucose (mmol/L)	8.95 (7.6–10.8)	5–18.3
Estimated HbA1c (%) (*n* = 124)	7.2 (6.4–8.4)	4.8–85
Estimated HbA1c (mmol/mol)	53 (44–68)	9.2–120
TAR (%)	37.5 (19.75–53)	0–97
TIR (%)	55 (39.75–72.25)	3–99
TBR (%)	5 (2–9)	0–33
Hypoglycemic events (*n*)	18 (9–49)	0–207
The average duration of hypoglycemic events (min)	96 (73.75–117.25)	0–207
isCGM data capture (%)	93 (83–97.5)	31–100
Scanning frequency	12 (8–15)	1–94

Intermittently scanned continuous glucose monitoring (isCGM), estimated glycated hemoglobin (eHbA1c), time above target range (TAR), time in target range (TIR), and time below target range (TBR).

**Table 5 healthcare-10-01792-t005:** Difference between baseline and estimated HbA1c.

	Median (Interquartile Range)		Difference ^†^	95% CI	*p* *
	Baseline (*n* = 121)	Estimated (*n* = 121)			
HbA1c	7.7 (6.97–8.83)	7.2 (6.4–8.4)	−0.35	−0.55 do −0.15	<0.001

* Wilcoxon test; glycated hemoglobin (eHbA1c), ^†^ Hodges-Lehmann median difference.

**Table 6 healthcare-10-01792-t006:** Differences in glycemic control among income groups.

	Number (%) of Patients by Income Level				*p* *
	<200 E	200–600 E	>600 E	Total	
Baseline HbA1c					
Good glycemic control	2 (11.1)	16 (25)	27 (38.6)	45 (29.6)	0.04
Poor glycemic control	16 (88.9)	48 (75)	43 (61.4)	107 (70.4)	
Total	18 (100)	64 (100)	70 (100)	152 (100)	
Estimated HbA1c					
Good glycemic control	3 (20)	26 (49.1)	31 (55.4)	60 (48.4)	0.05
Poor glycemic control	12 (80)	27 (50.9)	25 (44.6)	64 (51.6)	
Total	15 (100)	53 (100)	56 (100)	124 (100)	
TIR					
Good glycemic control	2 (12.5)	13 (24.5)	22 (38.6)	37 (29.4)	0.62
Poor glycemic control	14 (87.5)	40 (75.5)	35 (61.4)	89 (70.6)	
Total	16 (100)	53 (100)	57 (100)	126 (100)	

* *χ*^2^ test; glycated hemoglobin (HbA1c), time in target range (TIR), Euro (E, currency of the European Union).

**Table 7 healthcare-10-01792-t007:** Differences in glycemic control according to BMI.

	Number (%) of Patients			*p **
	Normal	Overweight	Total	
**Baseline HbA1c**				
Good glycemic control	33 (33)	12 (24)	45 (30)	0.24
Poor glycemic control	68 (67)	39 (76)	107 (70)	
Total	101 (100)	51 (100)	152 (100)	
**Estimated HbA1c**				
Good glycemic control	38 (47)	22 (51)	60 (48)	0.65
Poor glycemic control	43 (53)	21 (49)	64 (52)	
Total	81 (100)	43 (100)	124 (100)	
**TIR**				
Good glycemic control	24 (29)	13 (30)	37 (29)	0.97
Poor glycemic control	58 (71)	31 (70)	89 (71)	
Total	82 (100)	44 (100)	126 (100)	

* Chi-square Test; glycated hemoglobin (eHbA1c), time in target range (TIR).

**Table 8 healthcare-10-01792-t008:** Median and Cronbach alpha coefficient of internal consistency of reliability for each personality dimension.

	Median (Interquartile Range)	Cronbach Alpha
Extraversion	35 (30–39)	0.813
Agreeableness	40 (36–43)	0.688
Conscientiousness	41 (37–44)	0.757
Emotional stability	31 (26–36)	0.859
Intellect	35 (31–40)	0.780

**Table 9 healthcare-10-01792-t009:** Differences in personality dimensions related to asymptomatic hypoglycemia events.

	Median (Interquartile Range)		Difference ^†^	95% CI	*p* *
	Without Asymptomatic Hypoglycemia	With Asymptomatic Hypoglycemia			
Extraversion	35 (31–39)	36 (30–39)	0	−3 do 2	0.73
Agreeableness	39 (36–43)	42 (38–44)	2	0 do 4	0.03
Conscientiousness	40 (37–44)	41 (37–44)	0	−2 do 2	0.76
Emotional stability	32 (28–38)	30 (25–37)	−2	−5 do 1	0.24
Intellect	36 (31–40)	34 (33–39)	0	−2 do 2	0.75

CI—confidence interval; * Mann-Whitney U test; the ^†^ Hodges-Lehmann median difference.

**Table 10 healthcare-10-01792-t010:** Correlation of personality dimensions to Freestyle Libre glucose measurements and age.

	Spearman’s Rank Correlation Coefficient (*p*-Value)				
	Extraversion	Agreeableness	Conscientiousness	Emotional Stability	Intellect
TAR	0.016 (0.86)	−0.004 (0.96)	−0.031 (0.73)	−0.047 (0.60)	0.002 (0.99)
TIR	−0.006 (0.94)	0.004 (0.96)	−0.007 (0.94)	0.039 (0.66)	−0.003 (0.97)
TBR	−0.037 (0.68)	0.007 (0.94)	0.71 (0.06)	0.214 (0.02)	−0.058 (0.52)
Number of hypoglycemic events	−0.115 (0.24)	0.098 (0.32)	0.033 (0.74)	0.091 (0.35)	−0.003 (0.98)
The average duration of hypoglycemia (min)	0.125 (0.24)	−0.021 (0.84)	−0.054 (0.61)	−0.043 (0.69)	0.034 (0.75)
Sensor data captured (%)	−0.042 (0.67)	0.104 (0.29)	0.024 (0.81)	0.060 (0.54)	−0.068 (0.49)
Number of daily scans	−0.238 (0.009)	−0.030 (0.75)	−0.065 (0.48)	−0.069 (0.45)	−0.107 (0.24)
Average glucose	0.042 (0.65)	−0.039 (0.68)	−0.040 (0.67)	−0.100 (0.28)	0.005 (0.96)
Estimated HbA1c (%)	0.027 (0.76)	−0.035 (0.70)	−0.060 (0.51)	−0.080 (0.38)	−0.025 (0.78)
Estimated HbA1c (mmol/mol)	0.059 (0.52)	−0.086 (0.34)	−0.064 (0.48)	−0.053 (0.56)	0.001 (0.99)
Subject age	−0.304 (<0.001)	−0.106 (0.19)	−0.105 (0.20)	−0.178 (0.03)	−0.374 (<0.001)
Duration of diabetes	−0.068 (0.41)	0.016 (0.85)	−0.037 (0.65)	0.064 (0.43)	−0.185 (0.02)

Time above target range (TAR), time in target range (TIR), time below target range (TBR), and estimated glycated hemoglobin (eHbA1c).

**Table 11 healthcare-10-01792-t011:** Differences in personality dimensions related to the occurrence of complications.

	Median (Interquartile Range)		Difference ^†^	95% CI	*p* *
	Without Complications	With Chronic Complications			
Extraversion	36 (31–39)	35 (28–39)	−1	−4 do 1	0.22
Agreeableness	41 (37–43)	39 (36–43)	−1	−3 do 0	0.10
Conscientiousness	40 (36–44)	40 (37–44)	0	−2 do 2	0.98
Emotional stability	32 (27–36)	31 (25–37)	0	−3 do 2	0.85
Intellect	37 (33–40)	33 (30–39)	−2	−4 do 0	**0.02**

CI—confidence interval; * Mann-Whitney U test; the ^†^ Hodges-Lehmann median difference.

**Table 12 healthcare-10-01792-t012:** Differences in personality dimensions relating to the occurrence of polyneuropathy.

	Median (Interquartile Range)		Difference ^†^	95% CI	*p* *
	Without Polyneuropathy	With Polyneuropathy			
Extraversion	36 (31–40)	34 (28–39)	−2	−5 do 0	0.06
Agreeableness	40 (36–43)	39.5 (36–43)	0	−2 do 1	0.62
Conscientiousness	40 (37–44)	41 (36.75–44)	0	−2 do 2	0.84
Emotional stability	32 (26–38)	30 (25–36)	−1	−4 do 2	0.40
Intellect	37 (33–40)	33 (30–36)	−4	−5 do −2	<0.001

CI—confidence interval; * Mann-Whitney U test; the ^†^ Hodges-Lehmann median difference.

**Table 13 healthcare-10-01792-t013:** Correlation of duration of diabetes with baseline HbA1c and TIR (%).

	Spearman Correlation Coefficient *Rho* (*p* Value)
	Duration of Diabetes
Baseline HbA1c (%)	−0.015 (0.86)
Estimated HbA1c (%)	0.161 (0.08)
TIR (%)	−0.206 (0.02)

Glycated hemoglobin (HbA1c), time in target range (TIR).

## Data Availability

Not applicable.

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
