# Peer review of "The Association of Personality Traits and Parameters of Glycemic Regulation in Type 1 Diabetes Mellitus Patients Using isCGM"

_healthcare, 2022, doi:10.3390/healthcare10091792_

Round 1

Reviewer 1 Report

The manuscript of Sladic Rimac et al. describes the research performed to determinw influence of personality traits on glycemic regulation in T1D patients. Overall,  manuscript is well written, logical and structured. Some improvemnts still can be made.

1. The title. I would substitute the word "influence" with "association" or "relationship(s)". Influence is not optimal in this case. In the text changes may be made accordingly.

2. Last sentence in Abstract is a bit strange. I would rephrase it in the following direction: Pts. require closer follow-up/monitoring and should be instructed when neccessary... Please revise Conclusions  sectionin main text accordingly.

3.  Please note that modern CGM devices (e.g. Free Style Libre 3) may not require scanning and allerts indicating hypo or hyperglycemia may help patients to maintane safer glycemic control.

4. Although 7% HbA1c cut-off for "good" or "poor" glycemic control is generally acknowledged, there is more or less linear relationships of HbA1c and 6.9 will probably not differ much clinically from 7.1%. Additional correlation anylysis may be appropriate in this case.

5. text should be checked thoroughly for typos and English. Example: Table 3 Medijan; Abstract line 19  (BMI) WERE calculated - should be WAS etc.

6. Lines 258 to 321 is one paragraph. It should be split into reasonably sized paragraphs instead.

Author Response

Although ŠBT1 was traditionally considered a disease of thin people, obesity affects an increasing number of patients suffering from ŠBT1, the prevalence rate ranges between 2.8% and 37.1% [32], a high prevalence was also observed in this study (33.5%) No difference was observed among parameters of glycemic regulation and nutritional status, probably because the control of type 1 diabetes largely depends on regular self-control and adequate delivery of insulin, unlike type 2 diabetes.

Reviewer 2 Report

It is a very well written and informative manuscript. There are some minor mistakes I have found, for example, in line 163, Alpha = 0,05. Please go through all the manuscritp one more time. The introduction should have a strong scientific hypothesis behind the study.

Author Response

Review #2. It is a very well-written and informative manuscript.

  1. There are some minor mistakes I have found, for example, in line 163, Alpha = 0,05. Please go through all the manuscripts one more. Time

Thank you for your suggestion, we have checked the manuscript and typos have been corrected accordingly.

  1. The introduction should have a strong scientific hypothesis behind the study.

Thank you, we have corrected and clarified our hypothesis.

Reviewer 3 Report

Article

The influence of personality traits on glycemic regulation in Type 1 diabetes mellitus patients using isCGM

This study aimed to investigate how a patient's personality affects their ability to control their blood sugar levels in adults with Type 1 diabetes.

In deed, this study answered the research question but many comments are there.

-          It contains some of typing and grammatical mistakes

-          It is well written and designed

-          Objectives need some clarification to be clear.

-          How the authors determine the risk score of SARS-CoV-2 infection, do they have followed some guidelines, or they just followed previous studies, please provide reference.

General points to be considered

v  Title: perfect, but I suggested to add word parameters before glycemic regulation, to be      The influence of personality traits on parameters of glycemic regulation in Type 1 diabetes mellitus patients using isCGM

-          Abstract: perfect.  

1.      Introduction:

Well written

2.      Materials and Methods: well written, but authors have to add inclusion and exclusion criteria for the participants.

2.4 Statistical analysis: Well written

Results:

Well presented, but some notes are there:-

In Table 8. Differences in personality dimensions related to asymptomatic hypoglycemia events, in raw 3 the word Conscientiousnes s should be corrected (s should be removed).

-          I have not seen any comparison between the BMI with the levels of glucose or HBAIC, authors should add this comparison and discuss it.

Discussion: well written, but should be English language writing should be checked.

In line 314, space should be removed.

 and ultimately more prone to poor glycemic control [42],…… and the development of

Conclusion: well written, but could be improved.

v  References: citation of references should be according to journal instructions.

v  References should be checked and standardized.

Author Response

Review #3. This study aimed to investigate how a patient's personality affects the ability to control their blood sugar levels in adults with Type 1 diabetes.

 Indeed, this study answered the research question but many comments are there.

  1. It contains some typing and grammatical mistakes

Thank you for your suggestion, we have checked the manuscript and typos have been corrected accordingly.

  1. It is well-written and designed.

Thank you for your kind remark.

  1. Objectives need some clarification to be clear.

We have changed and clarified our objectives at the end of the Introduction section.

  1. How did the authors determine the risk score of SARS-CoV-2 infection, do they have followed some guidelines, or they just followed previous studies, please provide a reference.

We did not determine the risk score of SARS-CoV-2 infection, this was not the objective of our study, nor have we included that in our methodology.

General points to be considered

  1. Title: perfect, but I suggested adding word parameters before glycemic regulation, to be      The influence of personality traits on parameters of glycemic regulation in Type 1 diabetes mellitus patients using isCGM. Thank you for your suggestion, the Title has been changed accordingly.
  2. Abstract: perfect.  
  3. Introduction: Well written
  4. Materials and Methods: well written, but authors have to add inclusion and exclusion criteria for the participants.

Thanks for your suggestion, exclusion, and inclusion criteria were added in the Method section.

  1. Statistical analysis: Well written
  2. Results: Well presented, but some notes are there:-
  • In Table 8.Differences in personality dimensions related to asymptomatic hypoglycemia events, in raw 3 the word Conscientiousnes s should be corrected (s should be removed).

Thank you, we have corrected the sentence accordingly.

  • I have not seen any comparison between the BMI with the levels of glucose or HBAIC, authors should add this comparison and discuss it.
  • Thank you for this excellent point. Glycemic parameters according to BMI are presented in Table 7 in the Results section. Also, a paragraph was added in the Discussion section.
  1. Discussion: well written, but should be English language writing should be checked. In line 314, space should be removed and ultimately more prone to poor glycemic control [42],……and the development of

Thank you, changes have been made according to your suggestion.

  1. Conclusion: well written, but could be improved.

We have made further improvements in the Conclusion section.

  1. References: citation of references should be according to journal instructions. References should be checked and standardized.

We have checked and made the necessary correction in the reference section.